# The Transition from Pemphigus Foliaceus to Pemphigus Vegetans—An Intriguing Phenomenon within the Spectrum of Autoimmune Blistering Diseases: A Case Report

Olguța Anca Orzan [1,2,*] , Liliana Gabriela Popa [1,2,*], Iulia Badiu [1,2], Ana Ion [1,2] , Călin Giurcăneanu [1,2], Beatrice Bălăceanu-Gurău [1,2] and Irina Tudose [3]

1    Department of Oncologic Dermatology, Carol Davila University of Medicine and Pharmacy, 020021 Bucharest, Romania; dr.iulia.badiu@gmail.com (I.B.); anaion00@yahoo.com (A.I.); calin.giurcaneanu@umfcd.ro (C.G.); balaceanubeatrice@yahoo.com (B.B.-G.)
2    Department of Dermatology, Elias University Emergency Hospital, 011461 Bucharest, Romania
3    Department of Pathology, Elias University Emergency Hospital, 011461 Bucharest, Romania; irina_tds@yahoo.com
*    Correspondence: olguta.orzan@umfcd.ro (O.A.O.); lilidiaconu@yahoo.com (L.G.P.)

**Abstract:** Pemphigus vegetans and pemphigus foliaceus are rare autoimmune blistering diseases characterized by the disruption of desmosomal adhesion proteins, particularly desmoglein 3 and desmoglein 1. We report the case of a 62-year-old male who presented initially with scaly red plaques posing several diagnostic challenges. A histopathological examination revealed subcorneal acantholysis, matching the suspected clinical diagnosis of pemphigus foliaceus. The patient progressed, developing vegetating plaques, and a new biopsy was performed. The new histopathological and direct immunofluorescence exams were consistent with pemphigus vegetans. This case highlights the diagnostic challenges posed by the transition of pemphigus foliaceus to its vegetating form. We discuss the role of desmogleins in the pathogenesis of pemphigus and explore potential therapeutic strategies targeting these specific autoantigens.

**Keywords:** autoimmune blistering diseases; case report; desmoglein 1; desmoglein 3; pemphigus vegetans; pemphigus foliaceus

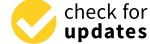



## 1. Introduction

Pemphigus encompasses a group of uncommon autoimmune dermatological conditions affecting the mucocutaneous membranes. In these conditions, the breakdown of cell-to-cell adhesion, known as acantholysis, leads to the formation of potentially life-threatening bullae and erosions [1].

Pemphigus foliaceus (PF) is one of several subtypes of this skin disease and has certain clinical and histopathological particularities, affecting exclusively the skin, without any mucosal lesions [2]. In PF, the regions affected the most are typically the head and neck, as well as the thoracic region [1]. Patients with PF have autoantibodies specifically targeting desmoglein 1, resulting in superficial blistering and cutaneous erosions, with the sparing of the mucosae in the majority of cases [3]. Upon clinical examination, blisters/bullae or erosions may be observed, although areas of erythema, scaling, and crusting are also frequently encountered [4]. In cases presenting with diffuse lesions, the differential diagnoses also include exfoliative eczema [4]. In the absence of mucosal lesions, a PF diagnosis could be confounded with one type of psoriasis or several types of eczema, especially seborrheic dermatitis.

Pemphigus vulgaris (PV) is the most frequently diagnosed subtype of this condition. PV's essential diagnostic feature is the presence of mucosal involvement, marked by flaccid blisters and erosions, primarily affecting the oropharynx [5]. While PV typically

manifests in the oral cavity, it can affect other mucosae including the ocular, esophageal, and genital mucosa [6]. The hallmark of this pemphigus subtype is the extreme discomfort caused by mucosal erosions, with significant nutritional consequences and protein–caloric malnutrition, ranging from mild to severe, and various degrees of weight loss [7].

A very rare subtype of PV is pemphigus vegetans, a loco-regional category of PV that generally involves flexures and fold-associated areas. However, lesions can be found on the skin covering any other part of the body. A typical lesion is usually a vegetating plaque with a cauliflower-like surface [8]. Similarly to PV, this subtype also presents with mucosal involvement [9]

The relationship between pemphigus and its genetic predisposing factors is an area of ongoing research. It is widely acknowledged that individuals with PV exhibit a heightened familial predisposition to other autoimmune disorders. Numerous investigations have elucidated specific associations between PV/PF and certain human leukocyte antigen (HLA) class II alleles [10]. For instance, studies have revealed a heightened prevalence of PV among the Ashkenazi Jewish population possessing the HLA-DRB1 0402 allele in comparison to other demographic groups [11]. Conversely, Lombardi et al. demonstrated that, in Italian patients, PV and PF share similar HLA alleles, suggesting additional factors (environment, drug exposure, dietary, etc.) that influence disease progression toward either PV or PF [12].

## 2. Case Report

We present a 62-year-old man evaluated in a dermatology clinic with a slightly pruritic, red, and scaly plaque lesion located on the skin in the right supraclavicular region. He noted the lesions 2 months ago. He had no personal or familial history of any skin disorders or autoimmune diseases. He denied taking any new medications or applying any skin products or deodorants prior to the appearance of this lesion. After several weeks, new lesions occurred on the trunk and a clinical diagnosis of pityriasis rosea was made. He was treated with topical corticosteroids (methylprednisolone acetate, a daily application for 3 weeks), but, despite this treatment, the lesions continued to enlarge and to spread to the whole body's skin surface. He was clinically reevaluated and a new diagnosis of psoriasis vulgaris was made. He was prescribed topical applications of clobetasol propionate 0.05% cream once daily for 4 months. However, the lesions continued to evolve and some transformed into many superficial blisters that were easily broken. The unroofed blisters were followed by scaly and crusted erosions (Figures 1 and 2). There was no mucosal involvement during the disease course.

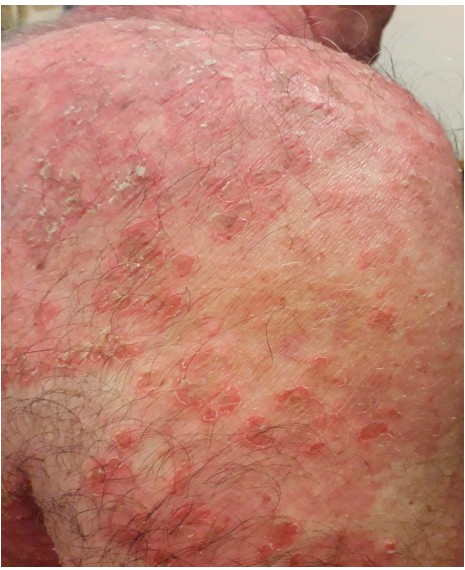

**Figure 1.** Small superficial blisters that evolve into erosions covered by crusts on the trunk.

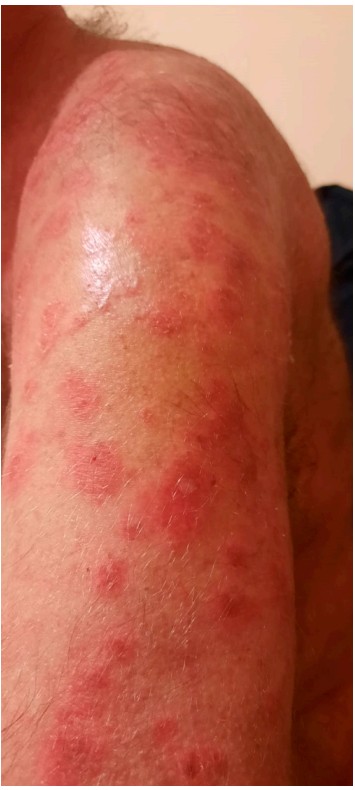

**Figure 2.** An intact superficial blister on the left arm.

Given the clinical aspect of these lesions, pemphigus foliaceus was suspected, and a skin biopsy was performed. Hematoxylin and eosin (H&E) staining showed sub-corneal acantholysis, confirming the clinically suspected diagnosis of pemphigus foliaceus (Figures 3 and 4). Direct immunofluorescence is mandatory for confirming the presence of intercellular immunoglobulin deposition. This testing was unable to be performed at that time due to limitations associated with the cost of this testing. Oral corticosteroid treatment with methylprednisolone was initiated at an initial dose of 0.8 mg/kg per day together with oral azathioprine at 50 mg per day. After one month, the lesions did not improve and the patient was referred back to our clinic for reevaluation (Figures 5–7).

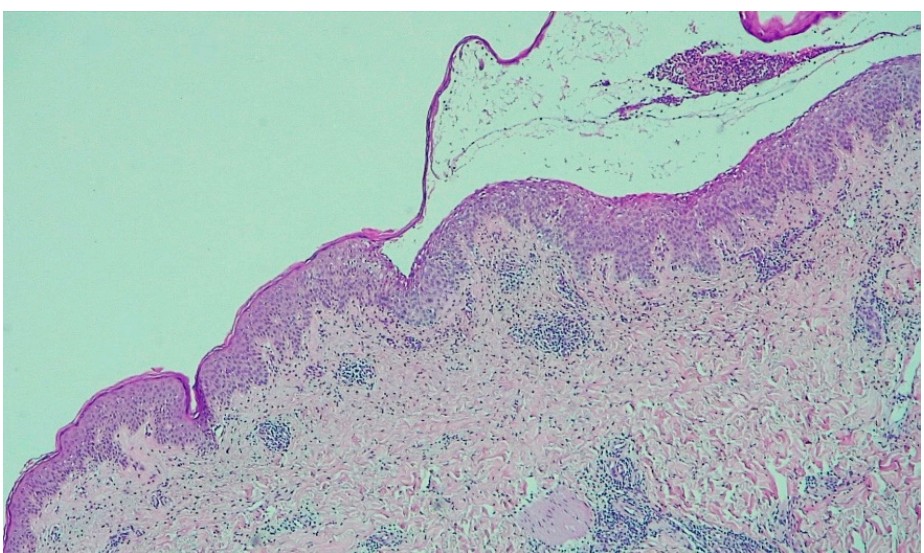

**Figure 3.** Subcorneal acantholysis consistent with the diagnosis of PF (H&E 50×).

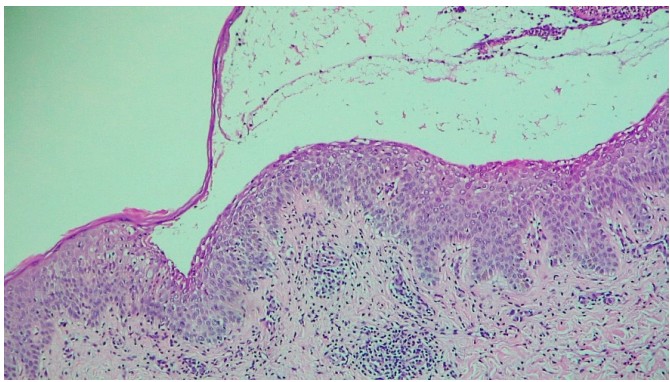

**Figure 4.** Subcorneal blister and acantholysis (H&E 100×).

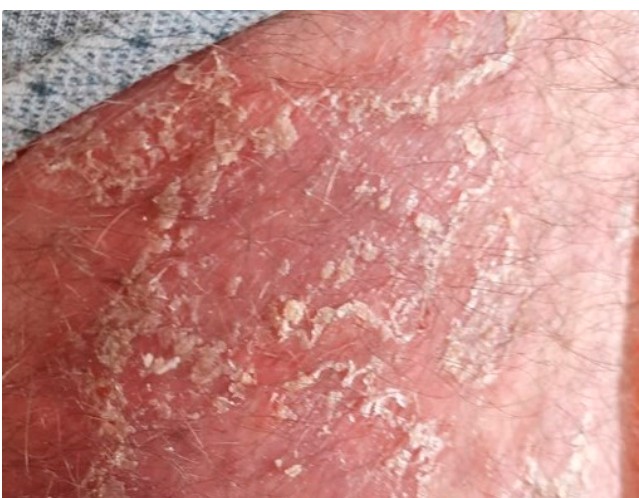

**Figure 5.** Exfoliative plaques.

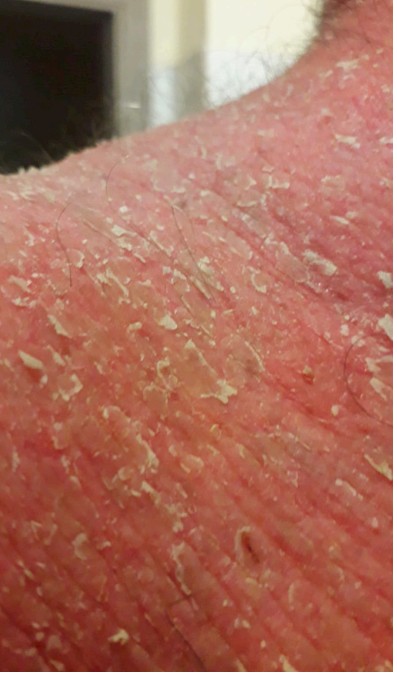

**Figure 6.** Exfoliative erythroderma after one month of therapy with methylprednisolone at 0.8 mg/kg per day associated with azathioprine at 50 mg per day.

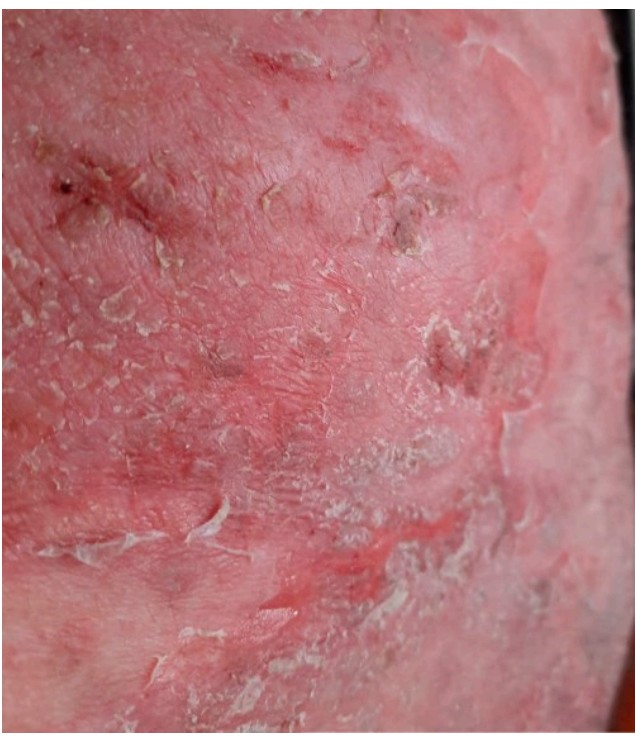

**Figure 7.** Exfoliative plaques covering large areas of the body.

Due to poor lesion control, azathioprine was changed to methotrexate at 15 mg per week, associated with folic acid supplementation at 5 mg three times per week and methyl-prednisolone at 64 mg per day. The corticosteroids were subsequently slowly tapered. This time, the lesions started and continued to improve (Figures 8 and 9).

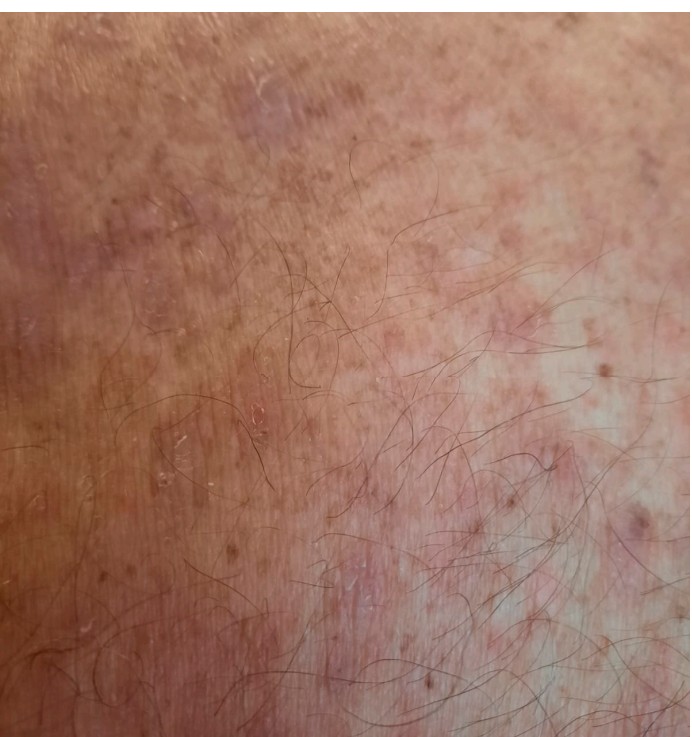

**Figure 8.** Healing of the erosions on treatment with methotrexate at 15 mg per week and with systemic corticosteroids tapered slowly.

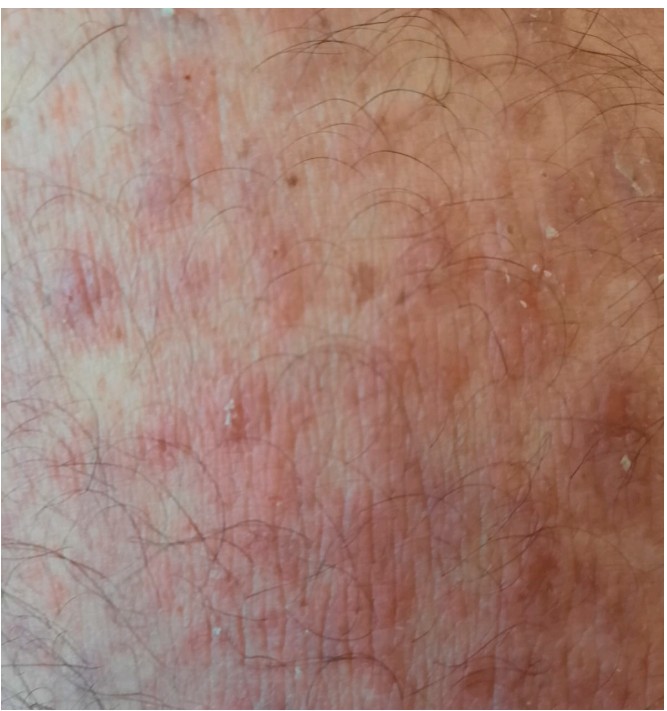

**Figure 9.** Healed erosions without the occurrence of new blisters.

After 2 months, the dose of methotrexate was decreased to 7.5 mg per week and methylprednisolone to 16 mg per day, and subsequently the patient developed new skin lesions consisting of flaccid blisters and painful erosions (Figure 10) accompanied by vegetating plaques in the neck area (Figure 11). Following this course, the doses of methotrexate and methylprednisolone were changed back to their initial starting doses, but this time the patient failed to show an improvement of his skin lesions.

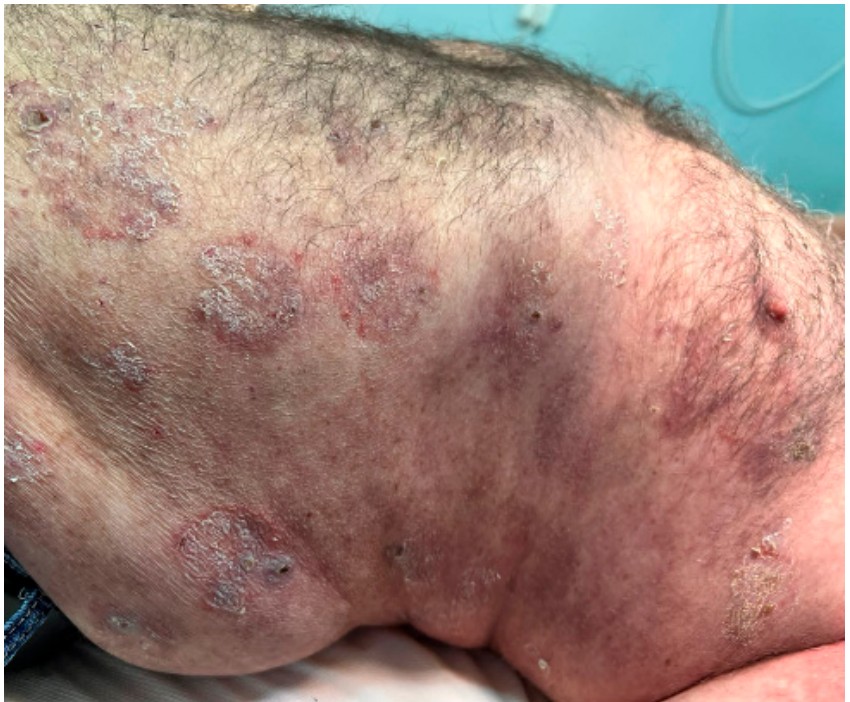

**Figure 10.** Large erosions scattered across the trunk.

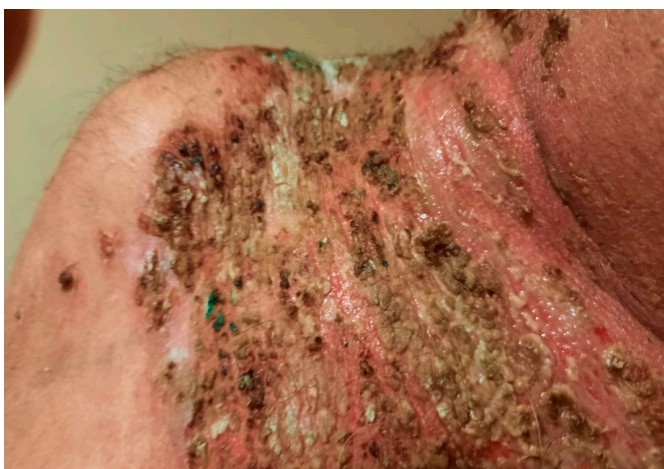

**Figure 11.** Vegetating plaques composed of granulation tissue and crusts in the neck area.

A new skin biopsy was performed that showed suprabasal acantholysis with hyperkeratosis, and papillomatosis with a downward proliferation of rete ridges (Figures 12 and 13).

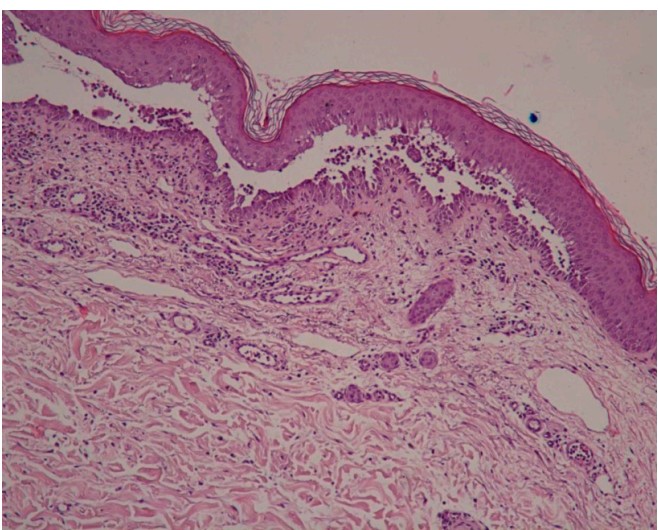

**Figure 12.** Suprabasal acantholysis with acanthosis and hyperkeratosis.

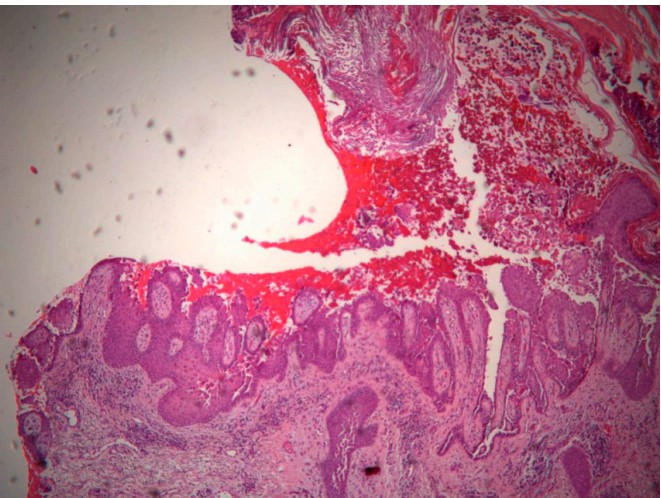

**Figure 13.** Downward proliferation of rete ridges.

A biopsy for a direct immunofluorescence exam was taken from a normal-appearing perilesional skin area and it showed an intercellular deposition of IgG (Figure 14).

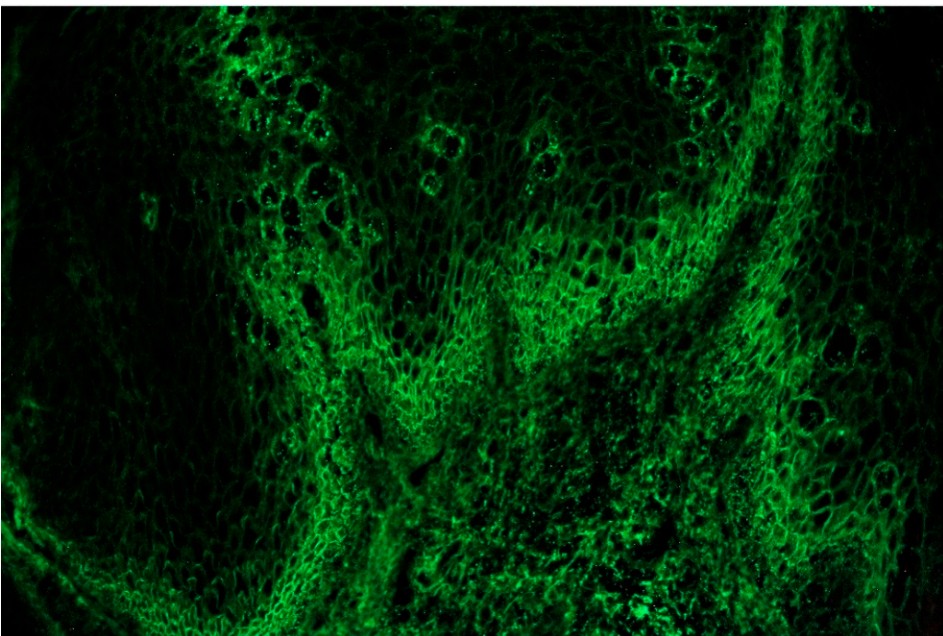

**Figure 14.** Direct immunofluorescence exam showed intercellular deposition of IgG.

Despite the sustained immunosuppression treatment with an increased dose of methyl-prednisolone (1 mg/ kg per day) and azathioprine at 100 mg per day, the patient's condition did not improve, and he developed new skin lesions (Figures 15 and 16) and oral mucosal lesions (Figure 17).

Due to the increased severity of the disease, rituximab was administered at a dose of 1 g every 2 weeks for 2 months, resulting in a spectacular improvement of the lesions.(Figures 18 and 19).

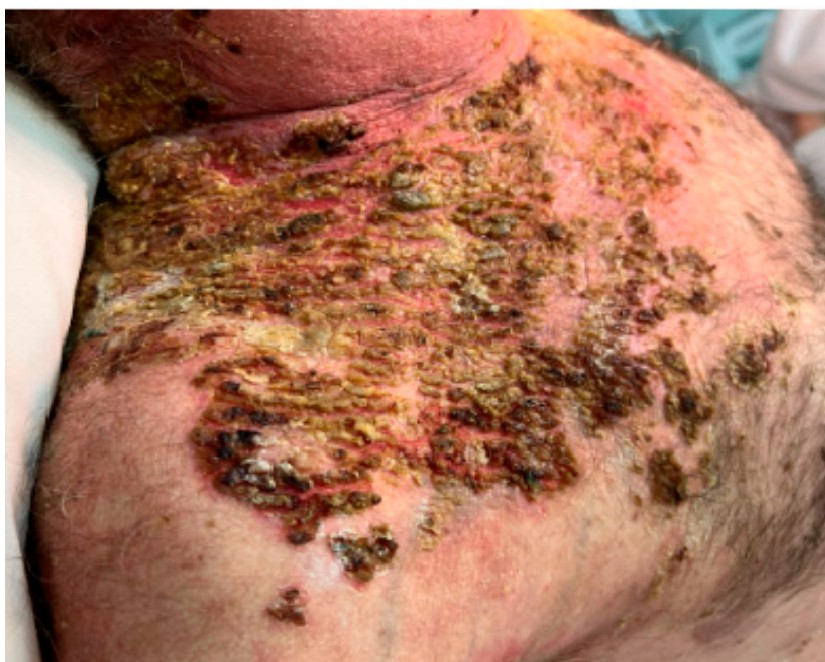

**Figure 15.** Before treatment with rituximab.

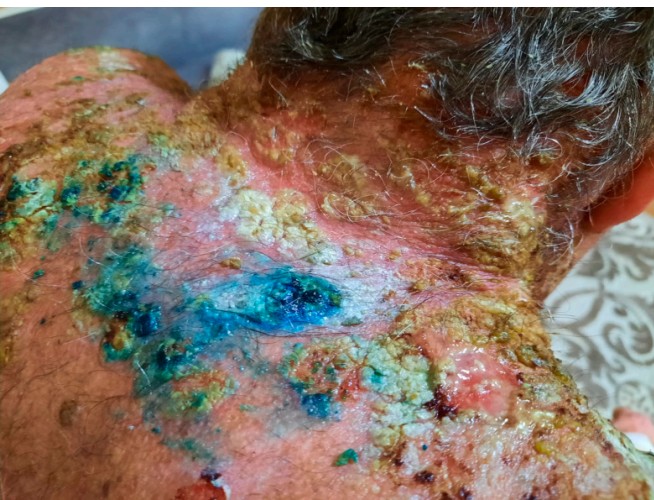

**Figure 16.** Severe vegetating plaques before rituximab treatment. The blue color is due to a methylene blue solution, applied for its antiseptic effect.

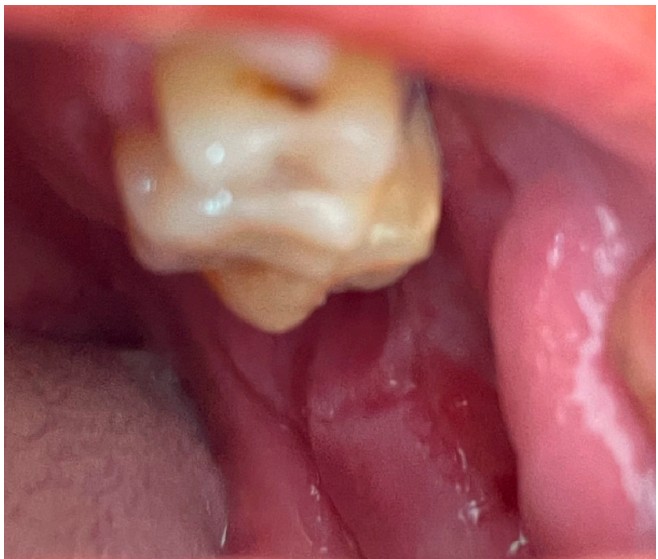

**Figure 17.** Multiple painful erosions on the buccal mucosa.

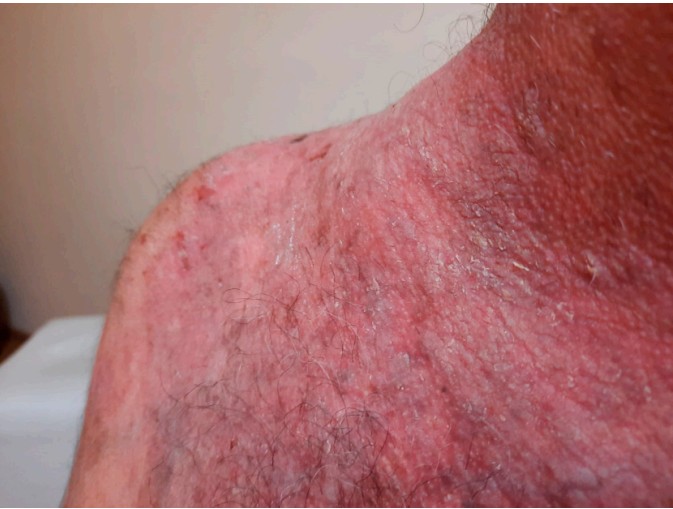

**Figure 18.** After the first two administrations of rituximab.

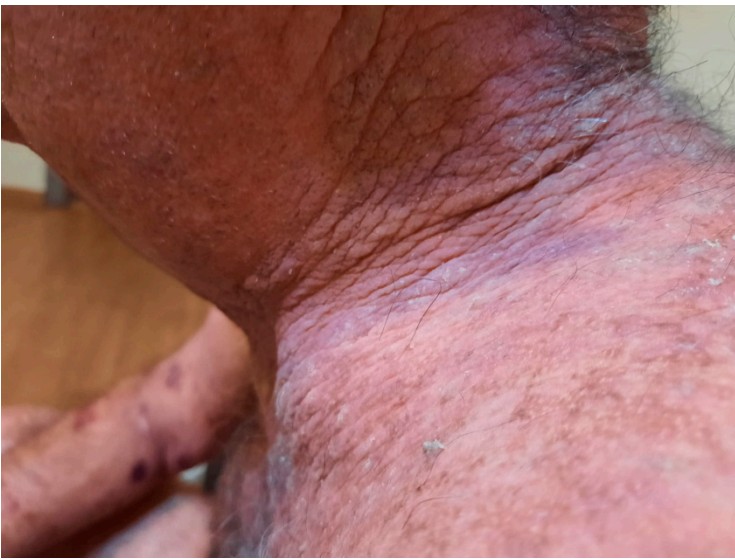

**Figure 19.** Improvement of skin lesions after the second dose of rituximab.

Based on the clinical appearance of the lesions, characterized by a vegetative nature, coupled with the results of the histopathological and direct immunofluorescence examinations, a diagnosis of pemphigus vegetans was established, explaining the more severe course and the need for more aggressive treatment. An additional test that could have further supported the diagnosis of pemphigus vegetans would have been the differentiated serum determination of anti-desmoglein 3 antibodies, which are found at higher levels in pemphigus vegetans compared to the foliaceus form, where anti-desmoglein 1 antibodies predominate. This test would have been useful in monitoring the response to treatment as well. Unfortunately, in this case, due to logistical reasons, it could not be performed; however, the rest of the investigations, combined with a clinical examination and the response to treatment, supported the diagnosis of pemphigus vegetans.

## 3. Discussions

Pemphigus vegetans represents the rarest clinical subtype of the various forms of pemphigus. It is different from pemphigus vulgaris as patients develop vegetative lesions occurring most frequently in intertriginous spaces [9]. The data in the literature describe two subtypes: the Hallopeau type, a milder variant that usually presents with initial pustules that transform into vegetative, pain-inducing lesions, and the Neumann subtype, a more aggressive presentation characterized by vesicles, bullae, and ulcerative lesions which in time become verrucous [10,11]. The case that we have presented correlates with the Neumann subtype, due to its clinical progress from large erosions to severe vegetating plaques composed of granulation tissue and crusts that covered the entire upper part of the trunk and the neck area. The first-line treatment of the majority of patients with pemphigus consists of a combination of corticosteroids and immunosuppressant medications. The lack of response to the first-line treatment was an indicator that we were dealing with a more severe category of disease requiring a different therapeutic approach [9,12,13].

The treatment with rituximab led to a very positive response in the case of our patient, with a significant improvement of his skin lesions. Even if in the past rituximab has been an off-label option for the treatment of patients with resistant pemphigus, recent evidence shows that the use of rituximab as a first-line treatment could be a safe treatment choice for managing patients with severe forms of pemphigus [14–17].

The particularity of our case lies in the rather peculiar transition from PF to PV. A potential immunological process possibly responsible for the shift from PF to PV involves epitope spreading, an initial inflammatory process that induces tissue damage by revealing concealed immunologic antigens, consequently triggering a subsequent autoimmune re-

sponse [18–21]. This progression might indicate that new autoantibodies emerge against desmoglein 3 after significant keratinocyte damage and the exposure of desmoglein 3, resulting in changes to the clinical presentation [22–24]. Data from the literature suggest that the progression of PV to PF is more prevalent than progressions from PF to PV [25–30].

## 4. Conclusions

While uncommon, the progression from PF to PV observed in this case underscores the significance of thorough follow-up care for patients with erythematous and bullous lesions. Further investigations into subtype transitions could uncover specific risk indicators and should help validate the mechanisms of this distinctive occurrence [29,30].

**Author Contributions:** Conceptualization, O.A.O. and L.G.P.; methodology, O.A.O.; software, I.B.; validation, O.A.O., C.G. and L.G.P.; formal analysis, I.T. and B.B.-G.; investigation, I.B. and L.G.P.; resources, L.G.P. and C.G.; data curation, B.B.-G. and I.B.; writing—original draft preparation, L.G.P., A.I. and O.A.O.; writing—review and editing, I.B., B.B.-G., L.G.P. and O.A.O.; visualization, A.I. and I.T.; supervision, O.A.O. and C.G.; project administration, L.G.P., I.B. and O.A.O. All authors have read and agreed to the published version of the manuscript.

**Funding:** This research received no external funding.

**Institutional Review Board Statement:** Not applicable.

**Informed Consent Statement:** Informed consent was obtained from the patient.

**Conflicts of Interest:** The authors declare no conflict of interest.

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
