# Peer review of "The Transition from Pemphigus Foliaceus to Pemphigus Vegetans—An Intriguing Phenomenon within the Spectrum of Autoimmune Blistering Diseases: A Case Report"

_dermato, doi:10.3390/dermato4020007_

Round 1
Reviewer 1 Report
Comments and Suggestions for Authors
The manuscript The Transition from Pemphigus Foliaceus to Pemphigus Vegetans – an Intriguing Phenomenon within the Spectrum of Autoimmune Blistering Diseases is interesting and touches on an important question. In general, I am inclined do recommend it for publication in “Dermato”.
I just have some minor comments and suggestions for the authors:
1. Did the authors follow any guidelines for preparing this case report?
Title
2. Please identify the report as a case report
Abstract
3. Please add an abstract explaining why this case is interesting and how it can expand the existing knowledge on the topic. Provide some details on clinical manifestations, interventions and outcomes.
1.Introduction
4. Clearly written and logically structured
Keywords
5. It is recommended to use MeSH terms and alphabetic order. I also suggest that the authors shoud add “case report” as a keyword.
2.Case report
6. A genetic component to the pathogenesis of pemphigus should be considered. Please provide relevant family and genetic information.
7. Line 106: “the development of new lesions on the skin and on the oral mucosal.” Please describe in details the oral manifestations. I suggest that the authors shoud add some photos of the oral lesions this patient developed.
3.Subheading number 3 is missing (“2. Case report” is followed by “4. Discussion”)
4.Discussion
8. Provides strengths and limitations of the approach used in this particular case and references for medical literature
Informed consent
9. Did the patient give his informed consent to publish this case report? Please provide this information in the manuscript
Comments on the Quality of English Language
Minor editing is required
Author Response
Please, find attached our answers. Thank you!

Reviewer 2 Report
Comments and Suggestions for Authors
Thank you for the opportunity to review your work. The case report describes an interesting case of Nuemann subtype pemphigus vegetans.
Comments on the Quality of English LanguageI didn't find any issues with the English quality in the paper.
Author Response
Thank you for your review. We have checked the English language and have made several editing corrections.